# Task-Agnostic Continual Reinforcement Learning: In Praise of a Simple Baseline

**Massimo Caccia**                                              *massimo.p.caccia@gmail.com*
*Amazon Web Services*
*Mila - Quebec AI Institute*
*Université de Montréal*

**Jonas Mueller**                                              *jonasmueller@csail.mit.edu*
*Cleanlab*

**Taesup Kim**                                                  *kim.ts.kr@gmail.com*
*Seoul National University*

**Laurent Charlin**                                            *lcharlin@gmail.com*
*Mila - Quebec AI Institute*
*HEC Montréal*
*Canada CIFAR AI Chair*

**Rasool Fakoor**                                              *fakoor@amazon.com*
*Amazon Web Services*

## Abstract

We study methods for task-agnostic continual reinforcement learning (TACRL). TACRL is a setting that combines the difficulties of *partially-observable* RL (a consequence of task agnosticism) and the difficulties of continual learning (CL), i.e., learning on a non-stationary sequence of tasks. We compare TACRL methods with their soft upper bounds prescribed by previous literature: multi-task learning (MTL) methods which do not have to deal with non-stationary data distributions, as well as task-aware methods, which are allowed to operate under *full observability*. We consider a previously unexplored and straightforward baseline for TACRL, replay-based recurrent RL (3RL), in which we augment an RL algorithm with recurrent mechanisms to mitigate partial observability and experience replay mechanisms for catastrophic forgetting in CL.

By studying empirical performance in a sequence of RL tasks, we find surprising occurrences of 3RL matching and overcoming the MTL and task-aware soft upper bounds. We lay out hypotheses that could explain this inflection point of continual and task-agnostic learning research. Our hypotheses are empirically tested in continuous control tasks via a large-scale study of the popular multi-task and continual learning benchmark Meta-World. By analyzing different training statistics including gradient conflict, we find evidence that 3RL's outperformance stems from its ability to quickly infer how new tasks relate with the previous ones, enabling forward transfer.

## 1 Introduction

Continual learning (CL) creates models and agents that can learn from a sequence of tasks. Continual learning agents promise to solve multiple tasks and adapt to new tasks without forgetting the previous one(s), a major limitation of standard deep learning agents (French, 1999; Thrun & Mitchell, 1995; McCloskey & Cohen, 1989; Lesort et al., 2020). In many studies, the performance of CL agents is compared against *multi-task*

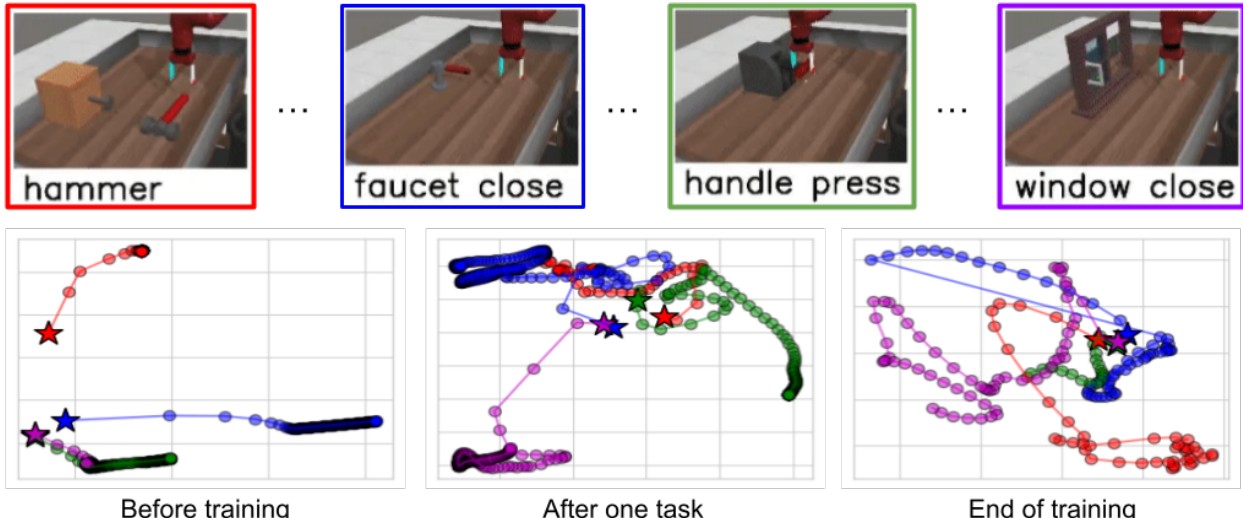

Figure 1: **The Continual-World**[1]**benchmark (top) as well as evolving RNN representations (bottom).**
Continual World consists of 10 robotic manipulation environments (four of which are shown above) within the same
state space and built on a common reward structure composed of shared components, i.e, reaching, grasping, placing
and pushing. We explore the task-agnostic setting in which agents need at least part of a trajectory (rather than a
single state) for task identification. We performed a PCA analysis of 3RL's RNN representations at different stages
of training. One episode is shown per task, and the initial task representation (at $t = 0$) is represented by a star. As
training progresses, the model learns i) a task-invariant initialization, drawing the initial states closer together, and
ii) richer, more diverse representations. Furthermore, the representations constantly evolve throughout the episodes,
suggesting the RNN performs more than task inference: it provides useful local information to the policy and critic
(see Hypothesis #3 in section 4).

(MTL) agents that are trained on all available tasks jointly. Further, during learning and evaluation, these
multi-task agents are typically provided with the identity of the current task (e.g. each datum is coupled with
its task ID). The performance of multi-task agents is used to provide a soft upper bound on the performance
of continual learning agents since the latter are restricted to learn from tasks in a given sequential order
that introduces new challenges, in particular *catastrophic forgetting* (McCloskey & Cohen, 1989). Moreover,
continual learning agents are often trained without knowing the task ID, a challenging setting motivated by
practical constraints and known as *task-agnostic* CL (Zeno et al., 2019; He et al., 2019; Caccia et al., 2020;
Berseth et al., 2021).

We find two observations challenging common beliefs in CL. First, we discover that TACRL agents endowed
with a recurrent memory can outperform task-aware agents. We report a second surprising discovery in these
agents, which are also augmented with the capability of replaying previous tasks' trajectories (Rolnick et al.,
2019), reaches the performance of its multi-task upper bound (as well as other multi-task RL methods),
despite only being exposed to the tasks in a sequential fashion. We refer to this methodology as *replay-based
recurrent reinforcement learning* (3RL) (see bottom of Figure 1 for a visualization of its representations).

We then pose several hypotheses to explain the underlying causes of these results. Guided by these hypotheses
we run a large-scale empirical study in which many RL methods are compared in different regimes. The
codebase to reproduce the results is available online[2]. Our study indicates that 3RL: 1) quickly infers how
new tasks relate to the previous ones, enabling forward transfer as well as 2) learns representations of the
underlying MDPs that reduce *task interference*, i.e., the situation when gradients of different tasks are in
conflict (Yu et al., 2020).

These findings question the need for forgetting-alleviating and task-inference tools for TACRL on a diverse
sequence of challenging and inter-related tasks (two properties that might be common in real-world appli-
cations). While it is conventional to assume that task-agnostic continual RL is strictly more difficult than

---

[2]https://github.com/amazon-research/replay-based-recurrent-rl

task-aware multi-task RL, this may not actually be the case for representative multi-task RL benchmarks like Meta-World. Despite being far more broadly applicable, TACRL methods may also perform just as well as their task-aware and multi-task counterparts.

## 2 Background & Task-agnostic Continual Reinforcement Learning

Here we formally define task-agnostic continual reinforcement learning (TACRL), and contrast it against multi-task RL as well as task-aware settings.

**MDP.** The RL problem is often formulated using a Markov decision process (MDP) (Puterman, 1994). An MDP is defined by the five-tuple $\langle \mathcal{S}, \mathcal{A}, \mathcal{T}, r, \gamma \rangle$ with $\mathcal{S}$ the state space, $\mathcal{A}$ the action space, $\mathcal{T}(s'|s,a)$ the transition probabilities, $r(s,a) \in \mathbb{R}$ or equivalently $r_i$ the reward obtained by taking action $a \in \mathcal{A}$ in state $s \in \mathcal{S}$, and $\gamma \in [0,1)$ a scalar constant that discounts future rewards. In RL, the transition probabilities and the rewards are typically unknown and the objective is to learn a policy, $\pi(a|s)$ that maximizes the sum of discounted rewards $\mathcal{R}_t^\pi = \sum_{i=t}^\infty \gamma^{i-t} r_i$ generated by taking a series of actions $a_t \sim \pi(\cdot|s_t)$. The Q-value $Q^\pi(s,a)$ corresponding to a policy $\pi$, is defined as the expected return starting at state $s$, taking $a$, and acting according to $\pi$ thereafter:

$$Q^\pi(s,a) = \mathbb{E}_\pi\Big[\sum_{t=0}^\infty \gamma^t r_t\Big] = r(s,a) + \gamma \mathbb{E}_{s',a'}\Big[Q^\pi(s',a')\Big]$$

**POMDP.** In most real world applications, if not all, the full information about an environment or a task is not always available to the agent due to various factors such as limited and/or noisy sensors, different states with identical observations, occluded objects, etc. (Littman et al., 1995; Fakoor & Huber, 2012). For this class of problems in which environment states are not fully observable by the agent, partially-observable Markov decision processes (POMDPs) Kaelbling et al. (1998) are used to model the problem. A POMDP is defined by a seven-tuple $\langle \mathcal{S}, \mathcal{A}, \mathcal{T}, \mathcal{X}, \mathcal{O}, r, \gamma \rangle$ that can be interpreted as an MDP augmented with an observation space $\mathcal{X}$ and a observation-emission function $\mathcal{O}(x'|s)$. In a POMDP, an agent cannot directly infer the current state of the environment $s_t$ from the current observation $x_t$. We split the state space into two distinct parts: the one that is observable $x_t$, which we refer to as $s_t^o$, and the remainder as the hidden state $s_t^h$, similarly to (Ni et al., 2021). To infer the correct hidden state, the agent has to take its history into account: the policy thus becomes $\pi(a_t|s_{1:t}^o, a_{1:t-1}, r_{1:t-1})$. An obvious choice to parameterize such a policy is with a recurrent neural network (Lin & Mitchell, 1993; Whitehead & Lin, 1995; Bakker, 2001; Fakoor et al., 2020b; Ni et al., 2021), as described in section 3. Like MDPs, the objective in POMDP is to learn a policy that maximizes the expected return $\mathbb{E}_{s^h}\Big[\mathbb{E}_\pi\big[\sum_{t=0}^\infty \gamma^t r_t\big]|s^h\Big]$.

**Task-agnostic Continual Reinforcement Learning (TACRL).** TACRL agents operate in a POMDP special case, explained next, designed to study the *catastrophic forgetting* that plagues neural networks (McCloskey & Cohen, 1989) when they learn on non-stationary data distributions, as well as *forward transfer* (Wolczyk et al., 2021a), i.e., a method's ability to leverage previously acquired knowledge to improve the learning of new tasks (Lopez-Paz & Ranzato, 2017). First, TACRL's environments assume that the agent does not have a causal effect on $s^h$. This assumption increases the tractability of the problem. It is referred to as an hidden-mode MDP (HM-MDP) (Choi et al., 2000). Table 1 provides its mathematical description.

The following assumptions help narrow down on the forgetting problem and knowledge accumulation abilities of neural networks. TACRL's assumes that $s^h$ follows a non-backtracking chain. Specifically, the hidden states are locally stationary and are never revisited. Finally, TACRL's canonical evaluation reports the anytime performance of the methods on all tasks, which we will refer to as *global return*. In this manner, we can tell precisely which algorithm have accumulated the most knowledge about all hidden states at the end of its *life*. In TACRL, the hidden states can be changed into a single categorical variable often referred to as

---

[2]The figures depict the rendering of Meta-World, and not what the agent observes. The agent's observation space is mainly composed of object, targets and gripper position. Because of the randomness of those positions, the agents needs more than one observation to properly infer the hidden state.

*context*, but more importantly in CRL literature, it represents a *task*. As each context can be reformulated as a specific MDP, we treat tasks and MDPs as interchangeable.

**Awareness of the Task Being Faced**   In practical scenarios, deployed agents cannot always assume full observability, i.e. to have access to a *task label* or *ID* indicating which task they are solving or analogously which hidden state they are in. They might not even have the luxury of being "told" when the task changes (unobserved task boundary): agents might have to infer it themselves in a data-driven way. We call this characteristic *task agnosticism* Zeno et al. (2019). Although impractical, CL research often treats *task-aware* methods, which observe the task label, as a soft upper bound to their task-agnostic counterpart (van de Ven & Tolias, 2019; Zeno et al., 2019). Augmented with task labels, the POMDP becomes fully observable, in other words it is an MDP.

**Multi-task Learning (MTL)**   For neural network agents, catastrophic forgetting can be simply explained by the stationary data distribution assumption of stochastic gradient descent being violated, such that the network parameters become specific to data from the most recent task. Thus it is generally preferable to train on data from all tasks jointly as in MTL (Zhang & Yang, 2021). However this may not be possible in many settings, and thus CL is typically viewed as a more broadly applicable methodology that is expected to perform worse than MTL (Rolnick et al., 2019; Chaudhry et al., 2018).

For RL specifically, multi-task RL (MTRL) often refers to scenarios with families of similar tasks (i.e. MDPs) where the goal is to learn a policy (which can be contextualized on each task's ID) that maximizes returns across *all* the tasks (Yang et al., 2020; Calandriello et al., 2014; Kirkpatrick et al., 2017a). While seemingly similar to CRL, the key difference is that MTRL assumes data from all tasks are readily available during training and each task can be visited as often as needed. These are often impractical requirements, which CRL methods are not limited by. Table 1 summarizes the settings we have discussed in this section. See App. A for a more thorough discussion on TACRL and its related settings.

| | T | $\pi$ | Objective | Evaluation |
|---|---|---|---|---|
| MDP Sutton & Barto (2018) | $p(s_{t+1}\|s_t,a_t)$ | $\pi(a_t\|s_t)$ | $\mathbb{E}_{\pi}\left[\sum_{t=0}^{\infty}\gamma^t r_t\right]$ | - |
| POMDP Kaelbling et al. (1998) | $p(s^h_{t+1},s^o_{t+1}\|s^h_t,s^o_t,a_t)$ | $\pi(a_t\|s^o_{1:t},a_{1:t-1},r_{1:t-1})$ | $\mathbb{E}_{s^h}\left[\mathbb{E}_{\pi}\left[\sum_{t=0}^{\infty}\gamma^t r_t\right]\|s^h\right]$ | - |
| HM-MDP Choi et al. (2000) | $p(s^o_{t+1}\|s^h_{t+1},s^o_t,a_t)p(s^h_{t+1}\|s^h_t)$ | $\pi(a_t\|s^o_{1:t},a_{1:t-1},r_{1:t-1})$ | $\mathbb{E}_{s^h}\left[\mathbb{E}_{\pi}\left[\sum_{t=0}^{\infty}\gamma^t r_t\right]\|s^h\right]$ | - |
| **Task-agnostic CRL** | $p(s^o_{t+1}\|s^h_{t+1},s^o_t,a_t)p(s^h_{t+1}\|s^h_t)$ | $\pi(a_t\|s^o_{1:t},a_{1:t-1},r_{1:t-1})$ | $\mathbb{E}_{s^h}\left[\mathbb{E}_{\pi}\left[\sum_{t=0}^{\infty}\gamma^t r_t\right]\|s^h\right]$ | $\mathbb{E}_{\tilde{s}^h}\left[\mathbb{E}_{\pi}\left[\sum_{t=0}^{\infty}\gamma^t r_t\right]\|s^h\right]$ |
| Task-Aware CRL | $p(s^o_{t+1}\|s^h_{t+1},s^o_t,a_t)p(s^h_{t+1}\|s^h_t)$ | $\pi(a_t\|s^h_t,s^o_t)$ | $\mathbb{E}_{s^h}\left[\mathbb{E}_{\pi}\left[\sum_{t=0}^{\infty}\gamma^t r_t\right]\|s^h\right]$ | $\mathbb{E}_{\tilde{s}^h}\left[\mathbb{E}_{\pi}\left[\sum_{t=0}^{\infty}\gamma^t r_t\right]\|s^h\right]$ |
| Multi-task RL | $p(s^o_{t+1}\|s^h_{t+1},s^o_t,a_t)p(s^h_{t+1})$ | $\pi(a_t\|s^h_t,s^o_t)$ | $\mathbb{E}_{\tilde{s}^h}\left[\mathbb{E}_{\pi}\left[\sum_{t=0}^{\infty}\gamma^t r_t\right]\|s^h\right]$ | - |

Table 1: Summarizing table of the settings relevant to TACRL. For readability purposes, $\tilde{s}^h$ denotes the stationary distribution of $s^h$. The Evaluation column if left blank when it is equivalent to the Objective one. The blue colorization highlights the changes occurring from one setting to the next.

## 3   Methods

In this section, we detail the base algorithm and different model architectures used for assembling different continual and multi-task learning baselines.

### 3.1   Algorithms

We use off-policy RL approaches which have two advantages for (task-agnostic) continual learning. First, they are more sample efficient than online-policy ones (Haarnoja et al., 2018a; Fakoor et al., 2020a). Learning from lower-data regimes is preferable for CRL since it is typical for agents to only spend short amounts of time in each task and for tasks to only be seen once. Second, task-agnostic CRL most likely requires some sort of replay function (Traoré et al., 2019; Lesort et al., 2019). This is in contrast to task-aware methods which can, at the expense of computational efficiency, *freeze-and-grow*, e.g. PackNet (Mallya & Lazebnik,

2018), to incur no forgetting. Off-policy methods, by decoupling the learning policy from the acting policy, support replaying of past data. In short, off-policy learning is the approach of choice in CRL.[3]

**Base algorithm**    The Soft Actor-Critic (SAC) (Haarnoja et al., 2018b) is an off-policy actor-critic algorithm for continuous actions. SAC adopts a maximum entropy framework that learns a stochastic policy which maximizes the expected return and also encourages the policy to contain some randomness. To accomplish this, SAC utilizes an actor/policy network $\pi_\phi$ and critic/Q network $Q_\theta$, parameterized by $\phi$ and $\theta$ respectively. Q-values are learnt by minimizing one-step temporal difference (TD) error by sampling previously collected data from the replay buffer (Lin, 1992). For more details on SAC, please look at App. B.

## 3.2  Models

We consider various architectures to handle multi-task learning (MTL) as well as continual learning (CL) in both task-aware and task-agnostic setting.

**Task ID modeling (TaskID).**    We assume that a model such as SAC can become task adaptive by providing task information to the networks. Task information such as task ID (e.g. one-hot representation), can be fed into the critics and actor networks as an additional input: $Q_\theta(s, a, \tau)$ and $\pi_\phi(a|s, \tau)$ where $\tau$ is the task ID. We refer to this baseline as Task ID modeling (TaskID). This method is applicable to both multi-task learning and continual learning.

**Multi-head modeling (MH).**    For multi-task learning (which is always task-aware), the standard SAC is typically extended to have multiple heads (Yang et al., 2020; Yu et al., 2019; Wolczyk et al., 2021b; Yu et al., 2020), where each head is responsible for a single distinctive task, i.e. $Q_\Theta = \{Q_{\theta_k}\}_k^K$ and $\pi_\Phi = \{\pi_{\phi_k}\}_k^K$ where $K$ denotes total number of tasks. MH is also applicable to all reinforcement learning algorithms. That way, the networks can be split into 2 parts: (1) a shared state representation network (feature extractor) and (2) multiple prediction networks (heads). This architecture can also be used for task-aware CL, where a new head is newly attached (initialized) when an unseen task is encountered during learning.

We also use this architecture in task-agnostic setting for both MTL and CL. Specifically, the number of heads is fixed a priori (we fix it to the number of total tasks) and the most confident actor head, w.r.t. the entropy of the policy, and most optimistic critic head are chosen. Task-agnostic multi-head (TAMH) can help us fraction the potential MH gains over the base algorithm: if MH and TAMH can improve performance, some of MH gains can be explained by its extra capacity instead of the additional task information.

**Task-agnostic recurrent modeling (RNN).**    Recurrent neural networks are able to encode the history of past data (Lin & Mitchell, 1993; Whitehead & Lin, 1995; Bakker, 2001; Fakoor et al., 2020b; Ni et al., 2021). Their encoding can implicitly identify different tasks (or MDP). Thus, we introduce RNNs as a history encoder where history is defined by $\{(s_i, a_i, r_i)\}_i^N$ and we utilize the hidden states $z$ as additional input data for the actor $\pi_\phi(a|s, z)$ and critic $Q_\theta(s, a, z)$ networks. This allows us to train models without any explicit task information, and therefore we use this modeling especially for task-agnostic continual learning. More details about the RNN are provided at the end of the next subsection.

## 3.3  Baselines

**FineTuning** is a simple approach to a CL problem. It learns each incoming task without any mechanism to prevent forgetting. Its performance on past tasks indicates how much forgetting is incurred in a specific CL scenario.

**Experience Replay (ER)** accumulates data from previous tasks in a buffer for retraining purposes, thus slowing down forgetting (Rolnick et al., 2019; Aljundi et al., 2019; Chaudhry et al., 2019; Lesort, 2020). Although simple, it is often a worthy adversary for CL methods. One limitation of replay is that, to

---

[3]Note that our findings are not limited to off-policy methods, in fact our 3RL model can be extended to any on-policy method as long as it utilizes a replay buffer (Fakoor et al., 2020a) Having the capability to support a replay buffer is more important than being on-policy or off-policy.

approximate the data distribution of all tasks, its compute requirements scale linearly with the number of tasks, leaving little compute for solving the current task, assuming a fixed compute budget. To achieve a better trade-off between remembering previous tasks and learning the current one, we use a strategy that caps replay by oversampling the data captured in the current task[4] explained in algorithm 1 L8-9.

**Multi-task (MTL)** trains on all tasks simultaneously and so it does not suffer from the challenges arising from learning on a non-stationary task distribution. It serves as a soft upper bound for CL methods.

**Independent** learns a set of separate models for each task whereby eliminating the CL challenges as well as the MTL ones, e.g., learning with conflicting gradients (Yu et al., 2020).

The aforementioned baselines are mixed-and-matched with the architectural choices to form different baselines, e.g. MTL with TaskID (MTL-TaskID) or FineTuning with MH (FineTuning-MH). At the core of this work lies a particular combination, explained next.

**Replay-based Recurrent RL (3RL)**  A general approach to TACRL is to combine ER—one of CL's most versatile baseline—with an RNN, one of RL's most straightforward approach to handling partial observability. We refer to this baseline as *replay-based recurrent RL* (3RL). As an episode unfolds, 3RL's RNN representations $z_t = \text{RNN}(\{(s_i, a_i, r_i)\}_{i=1}^{t-1})$ should predict the task with increasing accuracy, thus helping the actor $\pi_\theta(a|s, z)$ and critic $Q_\phi(s, a, z)$ in their respective approximations. We will see in section 4, however, that the RNN delivers more than expected: it enables forward transfer by decomposing new tasks and placing them in the context of previous ones. We provide pseudocode for 3RL in algorithm 1, which we kept agnostic to the base algorithm and not tied to episodic RL. Note that in our implementation, the actor and critics enjoy their own RNNs, as in Fakoor et al. (2020b); Ni et al. (2021): they are thus parameterize by $\theta$ and $\phi$, respectively. Our RNN implementation employs Gated Recurrent Units (GRU) (Chung et al., 2014), as prescribed by Fakoor et al. (2020b).

---

**Algorithm 1:** 3RL in TACRL

**Environment:** a set of $K$ MPDs, allowed timesteps $T$
**Input:** initial parameters $\theta$, empty replay buffers $\mathcal{D}$ and $\mathcal{D}^{\text{old}}$ , replay cap $\beta$, batch size $b$, history length $h$

**1  for** *task $\tau$ in $K$* **do**
**2**  $\quad$ set environment to $\tau^{th}$ MDP
**3**  $\quad$ **for** *times-steps $t$ in $T$* **do**
$\quad\quad\quad$ /* Sampling stage                                                                                                       */
**4**  $\quad\quad$ compute dynamic task representation $z_t = \text{RNN}_\theta(\{(s_i, a_i, r_i)\}_{i=t-h-1}^{t-1})$
**5**  $\quad\quad$ observe state $s_t$ and execute action $a_t \sim \pi_\theta(\cdot|s_t, z_t)$
**6**  $\quad\quad$ observe reward $r_t$ and next state $s_{t+1}$
**7**  $\quad\quad$ store $(s_t, a_t, r_t, s_{t+1})$ in buffer $\mathcal{D}$
$\quad\quad\quad$ /* Updating stage                                                                                                       */
**8**  $\quad\quad$ sample a batch $B$ of $b \times min(\frac{1}{n}, 1 - \beta)$ trajectories from the current replay buffer $\mathcal{D}$
**9**  $\quad\quad$ append to $B$ a batch of $b \times min(\frac{n-1}{n}, \beta)$ trajectories from the old buffer $\mathcal{D}^{\text{old}}$
**10**  $\quad\quad$ Compute loss on $B$ and accordingly update parameters $\theta$ with one step of gradient descent

---

## 4  Empirical Findings

We now present the empirical findings resulting from our task-agnostic continual RL (TACRL) experiments instantiated in challenging robotic manipulation tasks. Next, we investigate some alluring behaviours we have come upon, namely that replay-based recurrent reinforcement learning (3RL), a TACRL baseline, can outperform other task-agnostic but more importantly task-aware baselines, as well as match its MTL soft upper bound.

**Benchmarks**  The benchmark at the center of our empirical study is Meta-World (Yu et al., 2019), which has become the canonical evaluation protocol for multi-task reinforcement learning (MTRL) (Yu et al., 2020;

---

[4]note that to to this we maintain two separate buffers, which can only be done in a task-aware way. The desired behaviour can be achieved in task-agnostic way by simply oversampling recently collected data.

Yang et al., 2020; Kumar et al., 2020; Sodhani et al., 2021). Meta-World offers a suite of 50 distinct robotic manipulation environments. What differentiates Meta-World from previous MTRL and meta-reinforcement learning benchmarks (Rakelly et al., 2019) is the broadness of its task distribution. Specifically, the different manipulation tasks are instantiated in the same state and action space[5] and share a reward structure, i.e., the reward functions are combinations of reaching, grasping, and pushing different objects with varying shapes, joints and connectivity. Meta-World is thus fertile ground for algorithms to transfer skills across tasks, while representing the types of tasks likely relevant for real-world RL applications (see Figure 1 for a rendering of some of the environments). Consequently, its adoption in CRL is rapidly increasing (Wolczyk et al., 2021a; Mendez et al., 2020; Berseth et al., 2021). Benchmarking in this space is also an active research area (Mendez et al., 2020; Mendez & Eaton, 2022).

In this work, we study `CW10`, a benchmark introduced in Wolczyk et al. (2021a) with a particular focus on *forward transfer*, namely, by comparing a method's ability to outperform one trained from scratch on new tasks. `CW10` is composed of a particular subset of Meta-World conductive for forward transfer and prescribes 1M steps per task, where a step corresponds to a sample collection and an update. We selected `CW10` over `CW20`, which repeats the `CW10` twice, to save compute and thus perform a more extensive analysis.

We also study a new benchmark composed of the 20 first alphabetical tasks of Meta-World which we will use to explore a more challenging regime: the task sequence is twice as long and data and compute are constrained to half, i.e., 500k steps are allowed per task. We refer to this benchmark as `MW20`. We assembled all experiments with the Sequoia software (Normandin et al., 2022) for continual-learning research.

In terms of metrics, the reported global and current success are the average success on all tasks and average success on the task that the agent is currently learning, respectively.

**Experimental Details**   We use the hyperparameters prescribed by Meta-World for their Multi-task SAC (MTL-SAC) method. We ensured the performance of our SAC implementation on the `MT10`, one of Meta-World's prescribed MTRL benchmark, matches theirs (see App. E). The astute reader will find that: 1) our reported performances are much lower than those from the original Continual World, and 2) our baselines struggle with tasks learned easily in the single-task learning (STL) regime from the original Meta-World paper Yu et al. (2019). These discrepancies are explained by the fact that Wolczyk et al. (2021a) carried out their original Continual-World study using Meta-World v1, which is far from the updated v2 version we use here. For example, the state space is thrice larger in v2 and the reward functions have been completely rewritten. The original Meta-World study of Yu et al. (2019) was performed under far more generous data and compute settings (see App. E) and relied on MTL and STL specific architectures and hyperparameters.

For more details on our training procedures and hyperparameters, see App. D. We test the methods using 8 seeds and report 90% T-test confidence intervals as the shaded area of the figures. As explained in algorithm 1, we oversample recently gathered data. When oversampling, we set the replay cap at 80%, thus always spending at least 20% of the compute budget on the current task.

**Task Agnosticism overcomes Task Awareness**   Our first experiments are conducted on `CW10` and `MW20` and are reported in Figure 2. Interestingly, 3RL outperforms all other methods in both benchmarks. This is surprising for two reasons. First, at training time, the task-aware methods learn task-specific parameters to adapt to each individual task. Hence, they suffer less or no forgetting (in general). Second, at test time, the task-agnostic method has to spend time first inferring the task (see bottom of Figure 1 for a visualization), in contrast to task-aware methods.

Of course, one could add an RNN to a task-aware method. We intend to compare the effect of learning task-specific parameters compared to learning a common task inference network (the RNN). Nevertheless, a combination of the RNN with ER-MH which was unfruitful (see App. G).

To ensure that these results are not a consequence of the methods having different number of parameters, we learned the `CW10` benchmark with bigger networks and found the performance to drop across all methods (see App. H). Further, in our experiments ER-MH is the method with the most parameters and it is outperformed by 3RL.

---

[5]the fixed action space is an important distinction with traditional incremental supervised learning

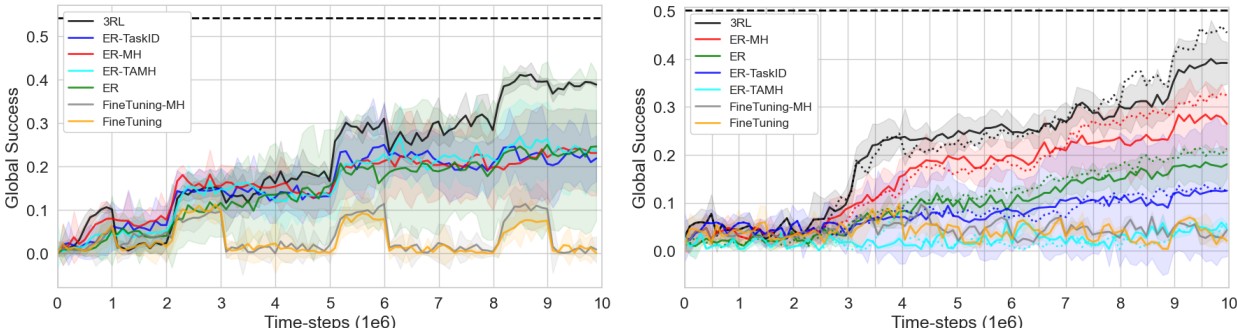

Figure 2: **3RL outperforms all baselines in both `CW10` (left) and `MW20` (right)**. The horizontal line is the reference performance of training independent models on all tasks. The dotted lines (right plot) represent the methods' performance when they oversample recently collected data. 3RL outperforms all other task-agnostic and more interestingly task-aware baselines. As a side note, ER is high variance as it attempts to solve the POMDP directly without having any explicit or implicit mechanism to do so.

**Continual Learning can Match Multi-Task Learning**  Multi-task learning is often used as a soft upper bound in evaluating CL methods in both supervised (Aljundi et al., 2019; Lopez-Paz & Ranzato, 2017; Delange et al., 2021) and reinforcement learning (Rolnick et al., 2019; Traoré et al., 2019; Wolczyk et al., 2021a). The main reason is that in the absence of additional constraints, multi-task learning (jointly training with data from all tasks in a stationary manner) does not suffer from the catastrophic forgetting that typically plagues neural networks trained on non-stationary data distributions.

3RL can reach this multi-task learning upper bound. In Figure 3 we report, for the second time, the results of the `MW20` experiments. This time, we focus on methods that oversample the current task and more importantly, we report the performance of each method's multi-task analog, i.e. their soft-upper bound (dotted line of the same color). Note that FineTuning methods do not have a MTRL counterpart and are thus not included in the current analysis. 3RL is the only approach that matches the performance of its MTRL equivalent. We believe it is the first time that a specific method achieves the same performance in a non-stationary task regime compared to the stationary one, amidst the introduced challenges like forgetting.

For the remainder of the section, we investigate some hypotheses that might explain 3RL's alluring behavior. We first hypothesise that the RNN boosts performance because it is simply better at learning a single MDP (Hypothesis #1). Next, we investigate the hypothesis that 3RL reduces parameter movement, as it is often a characteristic of successful continual learning methods (Hypothesis #2). We then explore the hypothesis that the RNN correctly places the new tasks in the context of previous ones (Hypothesis #3).

**Hypothesis #1: RNN individually improves the single-task performance**  A simple explanation is that the RNN enhances SAC's ability to learn each task independently, perhaps providing a different inductive bias beneficial to each individual task. To test this hypothesis, we run the `CW10` benchmark this time with each task learned separately, which we refer to as the Independent baselines. Note that this hypothesis is unlikely since the agent observes the complete state which is enough to act optimally (i.e. the environments are MDPs not POMDPs). Unsurprisingly, we find the RNN decreases the performance of standard SAC by 8.2% on average on all tasks. Accordingly, we discard this hypothesis. App. F provides the complete STL results.

**Hypothesis #2: RNN increases parameter stability, thus decreasing forgetting**  The plasticity-stability tradeoff is at the heart of continual learning: plasticity eases the learning of new tasks. Naive learning methods assume stationary data and so are too plastic in non-stationary regimes leading to catastrophic forgetting. To increase stability, multiple methods enforce (Mallya & Lazebnik, 2018) or regularize for (Kirkpatrick et al., 2017b) *parameter stability*, i.e., the tendency of a parameter to stay within its initial

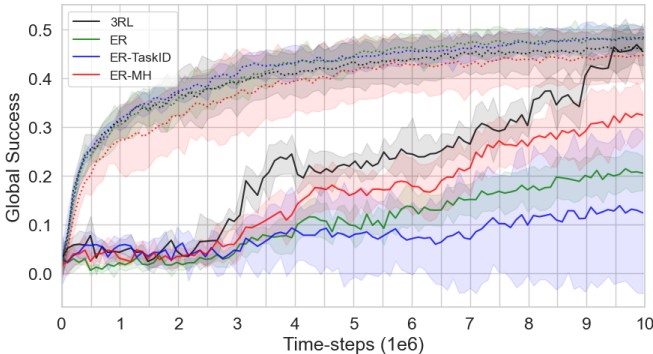

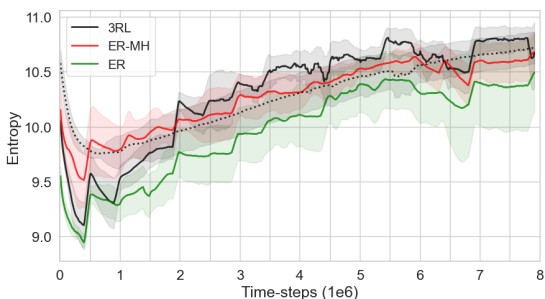

Figure 3: **3RL reaches its MTRL soft-upper bound**. Continual vs multi-task learning methods in solid lines and dotted lines, respectively. 3RL methods matches its soft-upper bound MTL analog as well as the other MTRL baselines. In contrast, other baselines' performance are drastically hindered by the non-stationary task distribution.

Figure 4: **3RL does not achieve superior continual learning performance through increased parameter stability**. We show the evolution of the methods' entropy in the parameters updates. We include MTL-RNN (dotted line) as a ref. We do not observe an increase in parameter stability: on the contrary, all methods, increasingly update more weights as new tasks (or data) come in.

value while new knowledge is incorporated. Carefully tuned task-aware methods, e.g. PackNet (Mallya & Lazebnik, 2018) in Wolczyk et al. (2021a), have the ability to prevent forgetting.[6]

Considering the above, we ask: could 3RL implicitly increase parameter stability? To test this hypothesis we measure the average total movement of the each weight throughout an epoch of learning which is defined by all updates in between an episode collection. To produce a metric suitable for comparing different runs which could operate in different regimes of parameter updates, we suppose report an entropy-like metric computed as follow $H_e = -(|\theta_{e+1} - \theta_e|)^\intercal \log(|\theta_{e+1} - \theta_e|)$, where $e$ is the epoch index. Note that the proposed metric takes all its sense when used relatively, (i.e. to compare the relative performance in-between methods) and not as an absolute measure of parameter stability Details about this experiment are found in App. I.

Figure 4 reports our proxy of total parameter movement for 3RL, ER, and ER-MH. We use these baselines since the gap between ER and its upper bound is the largest and the ER-MH gap is in between ER's and 3RL's. We find strong evidence to reject our hypothesis. After an initial increase in parameter stability, weight movement increases as training proceeds across all methods and even spikes when a new task is introduced (every 500K steps). MTL-RNN follows the same general pattern as the ER methods. Note that, it is not impossible that the *function* represented by the neural networks are stable even though their parameters are not. Nevertheless, it would be surprising that such a behaviour arose only in 3RL given the high similarity of the stability reported across all methods.

**Hypothesis #3: RNN increases computational efficiency through parameter sharing, thus improving performance in computationally-limited regimes** We examine the possibility that data and compute regimes scenarios we have experimented with are too challenging for methods with task-specific parameters like ER-MH and ER-TaskID. The task-aware methods learn task-specific parameters which might require more computation. In Figure 2 we compare the relative performance of the methods as we increase compute and data from 500k time-steps to 1M. It is important to note that although more data and compute generally increase the performance of multi-task learners, it is not necessarily the case for continual learners: these need to learn the current task while not forgetting the previous ones. This balancing act is often referred to as the stability-plasticity dilemma (Mermillod et al., 2013).

Nevertheless, we observe that increasing the data/compute narrows the gap between the 3RL and ER-MH. This increases the plausibility of hypothesis #3. However, some observations suggest this hypothesis is insufficient for explaining the phenomenon. First, in `CW10` (Figure 2) another high data and computational

---

[6]The observation that PackNet outperforms *an* MTRL baseline in Wolczyk et al. (2021a) is different from our stronger observation that a *single* method, namely 3RL, achieves the same performance in CRL than in MTRL

regime, the task-aware methods do struggle. Second, ER-TaskID, which has fewer extra parameters compared to ER-MH, does not work as well in general, and does not improve when data/compute is increased two-fold.

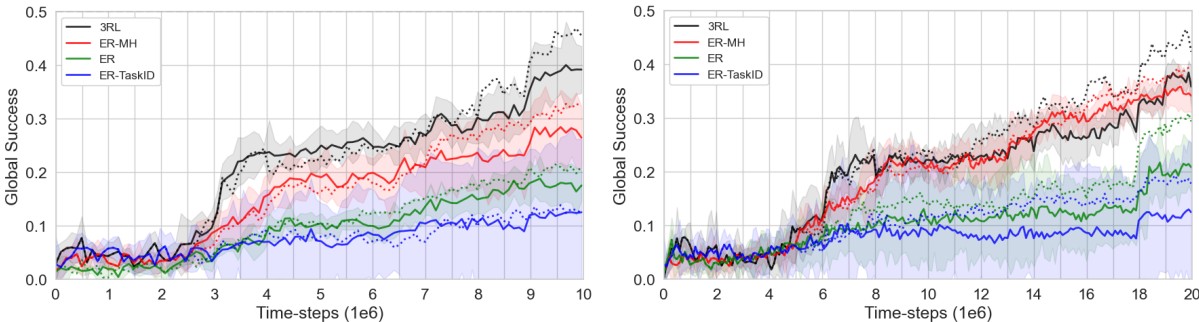

Figure 5: **Increased data/compute experiment.** We show some methods' performance on `MW20` with 500k steps (left) and 1M steps (right). More data and compute helped close the gap in one of the task-aware method (ER-MH) but not the other (ER-TaskID). Although there is some support for the hypothesis that 3RL outperformance is due to an increased computational efficiency, it is an insufficient explanation.

**Hypothesis #4: RNN correctly places the new tasks in the context of previous ones, enabling forward transfer and improving optimization** As in real robotic use-cases, MW tasks share a set of low-level reward components like grasping, pushing, placing, and reaching, as well as set of object with varying joints, shapes, and connectivity. As the agent experiences a new task, the RNN could quickly infer how the new data distribution relates with the previous ones and provide to the actor and critics a useful representation. Assume the following toy example: task one's goal is to grasp a door handle, and task two's to open a door. The RNN could infer from the state-action-reward trajectory that the second task is composed of two subtasks: the first one as well as a novel pulling one. Doing so would increase the policy learning's speed, or analogously enable forward transfer.

Now consider a third task in which the agent has to close a door. Again, the first part of the task consists in grasping the door handle. However, now the agent needs to subsequently push and not pull, as was required in task two. In this situation, task interference Yu et al. (2020) would occur. Once more, if the RNN could dynamically infer from the context when pushing or pulling is required, it could modulate the actor and critics to have different behaviors in each tasks thus reducing the interference. Note that a similar task interference reduction should be achieved by task-aware methods. E.g., a multi-head component can enable a method to take different actions in similar states depending on the tasks, thus reducing the task interference.

Observing and quantifying that 3RL learns a representation space in which the new tasks are correctly *decomposed* and placed within the previous ones is challenging. Our initial strategy is to look for effects that should arise if this hypothesis was true (so observing the effect would confirm the hypothesis).

First, we take a look at the time required to adapt to new tasks: if 3RL correctly infers how new tasks relate to previous ones, it might be able to learn faster by re-purposing learned behaviors. Figure 6 depicts the current performance of different methods throughout the learning of `CW10`. For reference, we provide the results of training separate models, which we refer to as Independent and Independent RNN.

The challenges of Meta-World v2 compounded with the ones from learning multiple policies in a shared network, and handling an extra level of non-stationary, i.e. in the task distribution, leaves the continual learners only learning task 0, 2, 5, and 8. On those tasks (except the first one in which no forward transfer can be achieved) 3RL is the fastest continual learner. Interestingly, 3RL showcases some forward transfer by learning faster than the Independent methods on those tasks. This outperformance is more impressive when we remember that 3RL is spending most of its compute replaying old tasks. We thus find some support for Hypothesis #3.

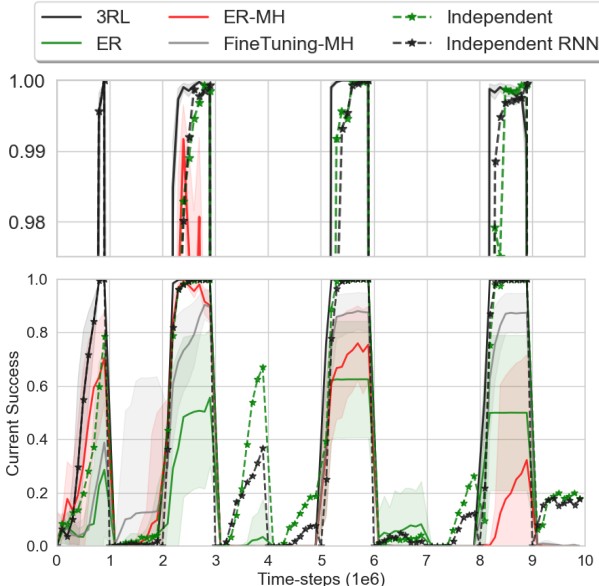

Figure 6: **3RL is the fastest continual learner.** Current success rate on `CW10`. The Independent method, which trains on each task individually, is still the best approach to maximise single-task performance. However, on the task that the continual learning methods succeed at, 3RL is the fastest learner. In these cases, its outperformance over Independent and Independent RNN indicates that forward transfer is achieved

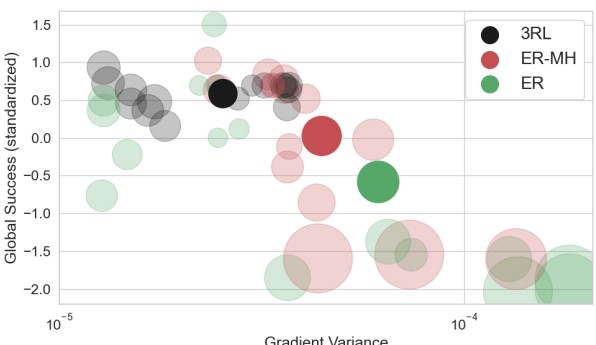

Figure 7: **3RL decreases gradient conflict leading to an increase in training stability and performance.** The global success and gradient as measured by the variance of the gradients are shown are plotted against each other. Training instability as measured by the variance of the Q-values throughout learning is represented by the markers' size, in a log-scale. Transparent markers depict seeds, whereas the opaque one the means. We observe a negative correlation between performance and gradient conflict (-0.75) as well as performance and training stability (-0.81), both significant under a 5% significance threshold. The hypothesis is that 3RL improves performance by reducing gradient conflict via dynamic task representations.

Second, simultaneously optimizing for multiple tasks can lead to conflicting gradients or task interference (Yu et al., 2020). To test for this effect, we use the average variance of the gradients on the mini-batch throughout the training as a proxy of gradient conflict. One might ask why we use the gradients' variance to measure gradient conflict. Yu et al. (2020) instead measure the conflict between two tasks via the angle between their gradients, proposing that the tasks conflict if this angle is obtuse. However we argue that it also critical to consider the magnitude of the gradients.

Consider an example of two tasks involving 1D optimization with two possible cases. In case 1, suppose that the gradient of the first task is 0.01 and the gradient of the second is -0.01. In case 2, suppose that the gradients are now 0.01 and 0.5, respectively. Measuring conflict via the angle would lead us to think that the tasks are in conflict in case 1 and are not in case 2. However, their agreement is actually much higher in case 1: both tasks agree that they should not move too far from the current parameter. In case 2, although the two tasks agree on the direction of the parameter update step, they have loss landscapes with great differences in curvature. The update step will be too small or too large for one of the tasks. We thus find variance to be a better measure of gradient conflict.

In Figure 7 we show the normalized global success metric plotted against the gradient variance. In line with our intuition, we do find that the RNN increases gradient agreement over baselines. As expected, adding a multi-head scheme can also help, to a lesser extent. We find a significant negative correlation of -0.75 between performance and gradient conflict. App. J reports the evolution of gradient conflict through time in the actor and critic networks.

Figure 7 also reports training stability, as measured by the standard deviation of Q-values throughout training (not to be confused with the parameter stability, at the center of Hypothesis #2, which measures how much the parameters move around during training). We find 3RL enjoys more stable training as well as a the significant negative correlation of -0.81 between performance and training stability. Note that The plausibility of Hypothesis #3 is thus further increased.

We wrap up the hypothesis with some qualitative support for it. Figure 1 showcases the RNN representations as training unfolds. If the RNN was merely performing task inference, we would observe the trajectories getting further from each other, not intersecting, and collapsing once the task is inferred. Contrarily, the different task trajectories constantly evolve and seem to intersect in particular ways. Although only qualitative, this observation supports the current hypothesis.

## 5 Related Work

To study CRL in realistic settings, Wolczyk et al. (2021b) introduce the Continual World benchmark and discover that many CRL methods that reduce forgetting lose their transfer capabilities in the process, i.e. that policies learned from scratch generally learn new tasks faster than continual learners. Previous works study CL and compare task-agnostic methods to their upper bounds (Zeno et al., 2019; van de Ven & Tolias, 2019) as well as CL methods compare to their multi-task upper bound (Ribeiro et al., 2019; Rolnick et al., 2019; Ammar et al., 2014). Refer to Khetarpal et al. (2020) for an in-depth review of continual RL as well as Lesort et al. (2021); Hadsell et al. (2020) for a CL in general.

Closer to our training regimes, TACRL is an actively studied field. Xu et al. (2020) uses a infinite mixture of Gaussian Processes to learn a task-agnostic policy. Kessler et al. (2021) learns multiples policies and casts policy retrieval as a multi-arm bandit problem. As for Berseth et al. (2021); Nagabandi et al. (2019), they use meta-learning to tackle the task-agnosticism part of the problem.

RNN were used in the context of continual supervised learning in the context of language modeling (Wolf et al., 2018; Wu et al., 2021) as well as in audio (Ehret et al., 2020; Wang et al., 2019). We refer to Cossu et al. (2021) for a in-depth review of RNN in continual supervised learning.

As in our work, RNN models have been effectively used as policy networks for reinforcement learning, especially in POMDPs where they can effectively aggregate information about states encountered over time (Wierstra et al., 2007; Fakoor et al., 2020b; Ni et al., 2021; Heess et al., 2015). RNN were used in the context of MTRL Nguyen & Obafemi-Ajayi (2019). Closer to our work, Sorokin & Burtsev (2019) leverages RNN in a task-aware way to tackle a continual RL problem. To the best of our knowledge, RNN have not been employed within TACRL nor combined with Experience Replay in the context of CRL.

## 6 Limitations

In this section we surface some limitations in our work. First of all, given the highly-demanding computational nature of the studied benchmarks, we could not run extensive hyperparameter searches as well as all the desired ablations. We have mostly relied on the hyperparameters prescribed by Meta-World, with the exception of the introduction of gradient clipping (App. D) which we found detrimental for the continual and multi-task learners to perform adequately. Importantly, we have not tested multiple context length for 3RL, an important hyperparameter for model-free recurrent RL (Ni et al., 2021). We can thus hypothesize that our 3RL's performances are underestimation.

Second, although Meta-World is know to be challenging, one could argue that its task-inference component is uncomplicated. Future work could explore task-agnostic continual RL benchmarks in which the different MDP are less distinctive.

Third, the hypothesis about the RNN correctly placing the new tasks in the context of previous ones (Hypothesis #3) is difficult to test. We have tested for effects that should arise if the hypothesis is true, but those effect could arise for different reasons. Future work could potentially dismantle the reward functions and look for RNN context overlaps across tasks when a particular reward component, e.g., grasping a door knob, is activated.

Lastly, we have only investigated a single benchmark. Introducing more benchmarks would was outside our compute budget. Moreover, challenging RL benchmarks have steep user learning curve, especially in continual RL. We leave for future work the study of 3RL in a wide suite of benchmarks.

# 7 Conclusion

We have shown that adding a recurrent memory to task-agnostic continual reinforcement learning allows TACRL methods like 3RL to match their multi-task upper bound and even outperform similar task-aware methods. Our large experiments suggest that 3RL manages to decompose the given task distribution into finer-grained subtasks that recur between different tasks, and that 3RL learns representations of the underlying MDP that reduce task interference.

Our findings question the conventional assumption that TACRL is strictly more difficult than task-aware multi-task RL. Despite being far more broadly applicable, TACRL methods like 3RL may nonetheless perform as well as their task-aware and multi-task counterparts. Although more work is required to come to a definitive conclusion, the need for the forgetting-alleviating and task-inference tools developed by the CL community for CRL is at now least questioned.

### Acknowledgments

We would like to thank Timothée Lesort and Lucas Caccia for providing valuable feedback. We also thank Samsung Electronics Co., Ldt. for their support.

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

# Appendix: Task-Agnostic Continual Reinforcement Learning: In Praise of a Simple Baseline

## A Extending TACRL's related settings

We continue the discussion on TACRL's related settings. In Meta-RL (Finn et al., 2017), the training task, or analogously the training hidden-states $s^h$, are different from the testing ones. We thus separate them into disjoint variable $s^h_{\text{train}}$ and $s^h_{\text{test}}$. In this setting, some fast adaptation to $s^h$ is always required. Meta-RL does not deal with a non-stationary training task distribution. Its continual counterpart however, i.e. Continual Meta-RL (Berseth et al., 2021), does. Table 2 summarizes the settings.

Noteworthy, the Hidden Parameter MDP (HiP-MDP) (Doshi-Velez & Konidaris, 2016) is similar to the HM-MDP but assumes a hidden states, i.e., the hidden states are samples i.i.d. Another setting similar to the HM-MD is the Dynamic Parameter MDP (Xie et al., 2020) in which the hidden-state are non-stationary but change at every episode.

| | T | $\pi$ | Objective | Evaluation |
|---|---|---|---|---|
| MDP (Sutton & Barto, 2018) | $p(s_{t+1}\|s_t, a_t)$ | $\pi(a_t\|s_t)$ | $\mathbb{E}_{\pi}\left[\sum_{t=0}^{\infty}\gamma^t r_t\right]$ | - |
| POMDP (Kaelbling et al., 1998) | $p(s^h_{t+1}, s^o_{t+1}\|s^h_t, s^o_t, a_t)$ | $\pi(a_t\|s^o_{1:t}, a_{1:t-1}, r_{1:t-1})$ | $\mathbb{E}_{s^h}\left[\mathbb{E}_{\pi}\left[\sum_{t=0}^{\infty}\gamma^t r_t\right]\|s^h\right]$ | - |
| HM-MDP (Choi et al., 2000) | $p(s^o_{t+1}\|s^h_{t+1}, s^o_t, a_t)p(s^h_{t+1}\|s^h_t)$ | $\pi(a_t\|s^o_{1:t}, a_{1:t-1}, r_{1:t-1})$ | $\mathbb{E}_{s^h}\left[\mathbb{E}_{\pi}\left[\sum_{t=0}^{\infty}\gamma^t r_t\right]\|s^h\right]$ | - |
| **Task-agnostic CRL** | $p(s^o_{t+1}\|s^h_{t+1}, s^o_t, a_t)p(s^h_{t+1}\|s^h_t)$ | $\pi(a_t\|s^o_{1:t}, a_{1:t-1}, r_{1:t-1})$ | $\mathbb{E}_{s^h}\left[\mathbb{E}_{\pi}\left[\sum_{t=0}^{\infty}\gamma^t r_t\right]\|s^h\right]$ | $\mathbb{E}_{\tilde{s}^h}\left[\mathbb{E}_{\pi}\left[\sum_{t=0}^{\infty}\gamma^t r_t\right]\|s^h\right]$ |
| Task-Aware CRL | $p(s^o_{t+1}\|s^h_{t+1}, s^o_t, a_t)p(s^h_{t+1}\|s^h_t)$ | $\pi(a_t\|s^h_t, s^o_t)$ | $\mathbb{E}_{s^h}\left[\mathbb{E}_{\pi}\left[\sum_{t=0}^{\infty}\gamma^t r_t\right]\|s^h\right]$ | $\mathbb{E}_{\tilde{s}h}\left[\mathbb{E}_{\pi}\left[\sum_{t=0}^{\infty}\gamma^t r_t\right]\|s^h\right]$ |
| Multi-task RL | $p(s^o_{t+1}\|s^h_{t+1}, s^o_t, a_t)p(s^h_{t+1})$ | $\pi(a_t\|s^h_t, s^o_t)$ | $\mathbb{E}_{\tilde{s}^h}\left[\mathbb{E}_{\pi}\left[\sum_{t=0}^{\infty}\gamma^t r_t\right]\|s^h\right]$ | - |
| Meta RL (Finn et al., 2017) | $p(s^o_{t+1}\|s^h_{t+1}, s^o_t, a_t)p(s^h_{t+1})$ | $\pi(a_t\|s^o_{1:t}, a_{1:t-1}, r_{1:t-1})$ | $\mathbb{E}_{\tilde{s}^h_{\text{train}}}\left[\mathbb{E}_{\pi}\left[\sum_{t=0}^{\infty}\gamma^t r_t\right]\|s^h_{\text{train}}\right]$ | $\mathbb{E}_{\tilde{s}^h_{\text{test}}}\left[\mathbb{E}_{\pi}\left[\sum_{t=0}^{\infty}\gamma^t r_t\right]\|s^h_{\text{test}}\right]$ |
| Continual Meta-RL | $p(s^o_{t+1}\|s^h_{t+1}, s^o_t, a_t)p(s^h_{t+1}\|s^h_t)$ | $\pi(a_t\|s^o_{1:t}, a_{1:t-1}, r_{1:t-1})$ | $\mathbb{E}_{s^h_{\text{train}}}\left[\mathbb{E}_{\pi}\left[\sum_{t=0}^{\infty}\gamma^t r_t\right]\|s^h_{\text{train}}\right]$ | $\mathbb{E}_{\tilde{s}^h_{\text{test}}}\left[\mathbb{E}_{\pi}\left[\sum_{t=0}^{\infty}\gamma^t r_t\right]\|s^h_{\text{test}}\right]$ |

Table 2: Summarizing table of the settings relevant to TACRL. For readability purposes, $\tilde{s}^h$ denotes the stationary distribution of $s^h$. The Evaluation column if left blank when it is equivalent to the Objective one.

## B Soft-Actor Critic

The Soft Actor-Critic (SAC) (Haarnoja et al., 2018b) is an off-policy actor-critic algorithm for continuous actions. SAC adopts a maximum entropy framework that learns a stochastic policy which not only maximizes the expected return but also encourages the policy to contain some randomness. To accomplish this, SAC utilizes an actor/policy network (i.e. $\pi_\phi$) and critic/Q network (i.e. $Q_\theta$), parameterized by $\phi$ and $\theta$ respectively. Q-values are learnt by minimizing one-step temporal difference (TD) error by sampling previously collected data from the replay buffer (Lin, 1992) denoted by $\mathcal{D}$.

$$\mathcal{J}_Q(\theta) = \mathbb{E}_{s,a}\left[\left(Q_\theta(s,a) - y(s,a)\right)^2\right], \ a' \sim \pi_\phi(\cdot|s') \tag{1}$$

where $y(s,a)$ is defined as follows:

$$y(s,a) = r(s,a) + \gamma \mathbb{E}_{s',a'}\left[Q_{\hat{\theta}}(s',a') - \alpha \log(a'|s')\right]$$

And then, the policy is updated by maximizing the likelihood of actions with higher Q-values:

$$\mathcal{J}_\pi(\phi) = \mathbb{E}_{s,\hat{a}}\left[Q_\theta(s,\hat{a}) - \alpha \log \pi_\phi(\hat{a}|s)\right], \ \hat{a} \sim \pi_\phi(\cdot|s) \tag{2}$$

where $(s, a, s') \sim \mathcal{D}$ (in both equation 1 and equation 2) and $\alpha$ is entropy coefficient. Note that although SAC is used in this paper, other off-policy methods for continuous control can be equally utilized for CRL. SAC is selected here as it has a straightforward implementation and few hyper-parameters.

## C Baselines Definitions

**FineTuning-MH** is FineTuning with task-specific heads. For each new task, it spawns and attaches an additional output head to the actor and critics. Since each head is trained on a single task, this baseline allows to decompose forgetting happening in the representation of the model (trunk) compared to forgetting in the last prediction layer (head). It is a task-aware method.

**ER-TaskID** is a variant of ER that is provided with task labels as inputs (i.e. each observation also contains a task label). It is a task-aware method that has the ability to learn a task representation in the first layer(s) of the model.

**ER-MH** is ER strategy which spawns tasks-specific heads (Wolczyk et al., 2021a), similar to FineTuning-MH. ER-MH is often the hardest to beat task-aware baseline (). ER-TaskID and ER-MH use two different strategies for modelling task labels. Whereas, MH uses $|h| \times |A|$ task-specific parameters (head) taskID only uses $|h|$, with $|h|$ the size of the network's hidden space (assuming the hidden spaces at each layer have the same size) and $|A|$ the number of actions available to the agent.

**ER-TAMH (task-agnostic multi-head)** is similar to ER-MH, but the task-specific prediction heads are chosen in a task-agnostic way. Specifically, the number of heads is fixed a priori (in the experiments we fix it to the number of total tasks) and the most confident actor head, w.r.t. the entropy of the policy, and most optimistic critic head are chosen. ER-TAMH has the potential to outperform ER, another task-agnostic baseline, if it can correctly infer the tasks from the observations.

**MTL** is our backbone algorithm, namely SAC, trained via multi-task learning. It is the analog of ER.

**MTL-TaskID** is MTL, but the task label is provided to the actor and critic. It is the analog of ER-TaskID and is a standard method, e.g. (Haarnoja et al., 2018a).

**MTL-MH** is MTL with a task-specific prediction network. It is the analog of ER-MH and is also standard, e.g. (Yu et al., 2020; Haarnoja et al., 2018a; Yu et al., 2019).

**MTL-TAMH** is similar to MTL-MH, but the task-specific prediction heads are chosen in the same way as in ER-TAMH.

**MTL-RNN** is similar to MTL, but the actor and critic are mounted with an RNN. It is the analog of 3RL.

## D Experimental Details

### D.1 Hyperparameters and their justification

The choice of hyperparameters required quite some work. Initially, we used the SAC hyperparameters prescribed by Continual World (Wolczyk et al., 2021a), designed on Meta-World v1, without any success. We then tried some of Meta-World v2's prescribed hyperparameter, which helped us match MetaWorld's multi-task reported results.

However, the continual and multi-task learning baselines would still suffer from largely unstable training due to the deadly triad (van Hasselt et al., 2018) problems in CRL and MTRL. After further experimentation, we observed that gradient clipping could stabilize training, and that clipping the gradients to a norm of 1 achieved the desired behavior across all methods, except for the multi-head baseline in which 10 was more appropriate.

Lastly, we use automatic entropy tuning except in the MTRL experiments, where we found its omission to be detrimental. Because their MT-SAC implementation learns a task-specific entropy term, we think this is the reason why Yu et al. (2019) do not observe the same behavior. All hyperparameters are summarized in Table 3.

| | | | |
|---|---|---|---|
| Architecture | | 2-layer MLP | |
| hidden state | | [400, 400] | |
| activation | | ReLU | |
| episode length | | Continual World: 200, else: 500 | |
| minimum buffer size | | 10 tasks: 1500, 20 tasks: 7500 | |
| batch size | | | |
| learning rate | | $1 \times 10^{-3}$ | |
| soft-target interpolation | | $5 \times 10^{-3}$ | |
| RNN's context length | | 15 | |
| burn in period | | 10,000 steps | |
| | **Independent** | **MTRL** | **CRL** |
| automatic-entropy tuning | on | off | on |
| gradient clipping | None | MH: 10, else: 1 | MH: 10, else: 1 |

Table 3: **Table of hyperparameters.** The top hyperparameters are global, whereas the bottom ones are setting specific.

## D.2 Computing Resources

All experiments were performed on Amazon EC2's P2 instances which incorporates up to 16 NVIDIA Tesla K80 Accelerators and is equipped with Intel Xeon 2.30GHz cpu family.

All experiments included in the paper can be reproduced by running 43 method/setting configurations with 8 seeds, each running for 4.2 days on average.

## D.3 Software and Libraries

In the codebase we've used to run the experiments, we have leverage some important libraries and software. We used Mujoco (Todorov et al., 2012) and Meta-World (Yu et al., 2019) to run the benchmarks. We used Sequoia (Normandin et al., 2022) to assemble the particular CRL benchmarks, including `CW10`. We used Pytorch (Paszke et al., 2019) to design the neural networks.

# E   Validating our SAC implementation on MT10

In Figure 8 we validate our SAC implementation on Meta-World v2's MT10.

# F   Extended CW10 Single-task results

Results for single-task experiments are shown in figure 9

# G   Task-aware meets task-agnostic

In figure 10 we show that combining the RNN with MH is not a good proposition in `CW10`.

# H   Larger networks do not improve performance

In figure 11 we report that increasing the neural net capacity does not increase performance.

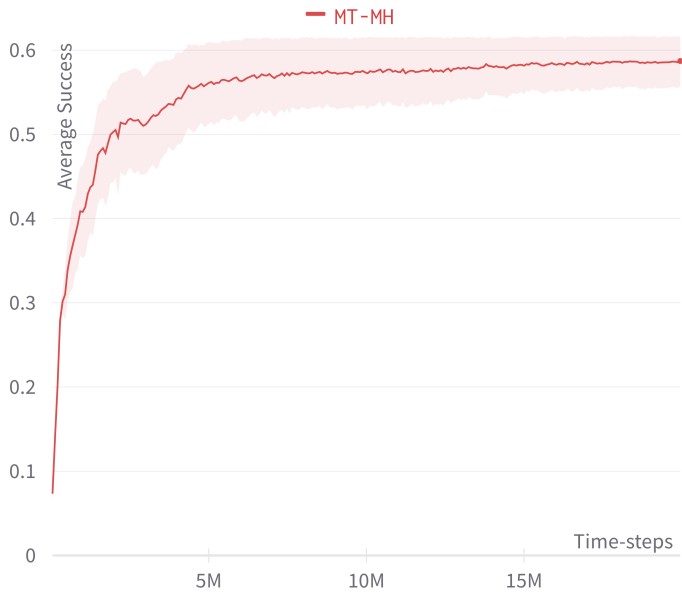

Figure 8: **MT10 experiment**. We repeat the popular MT10 benchmark with our MT-MH implementation. After 20M time-steps, the algorithm reaches a success rate of 58%. This is in line with Meta-World reported results. In Figure 15 of their Appendix, their MT-SAC is trained for 200M time-steps the first 20M time-steps are aligned with our curve. We further note that our CW10 result might seem weak vis à vis the reported ones in Wolczyk et al. (2021b). This is explained by Wolczyk et al. (2021b) using Meta-World-v1 instead of the more recent Meta-World-v2.

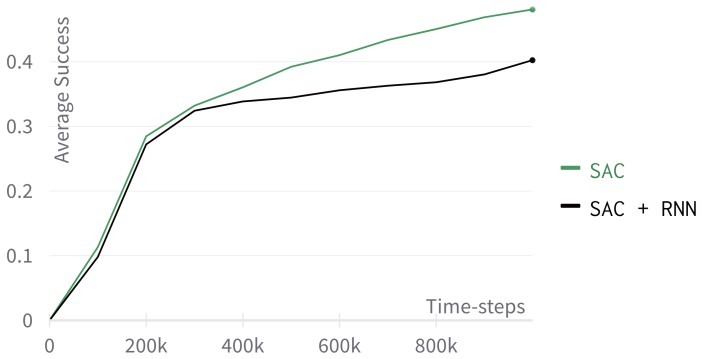

Figure 9: **Single-task learning CW10 experiments**. Average success on all task trained independently. In this regime, the RNN does not help.

## I  Gradient Entropy Experiment

To assess parameter stability, we look at the entropy of the parameters for each epoch. Because the episodes are of size 500, the epoch corresponds to 500 updates. To remove the effect of ADAM (Kingma & Ba, 2017), our optimizer, we approximate the parameters' movement by summing up their absolute gradients throughout the epoch. To approximate the sparsity of the updates, we report the entropy of the absolute gradient sum. For example, a maximum entropy would indicate all parameters are moving equally. If the entropy drops, it means the algorithm is applying sparser updates to the model, similarly to PackNet.

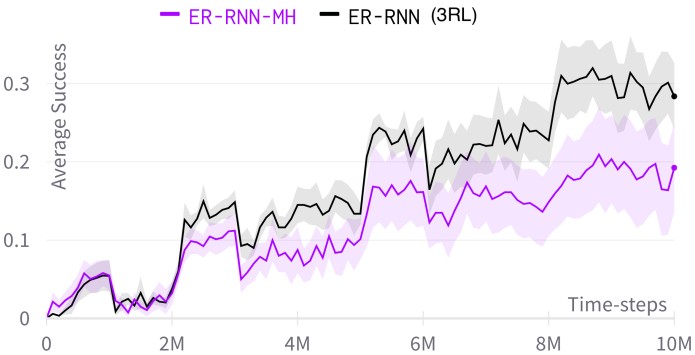

Figure 10: **CW10 experiment combining the RNN and MH**. Combining the best task-agnostic (3RL) and task-aware (ER-MH) CRL methods did not prove useful. Note that this experiment was ran before we enstored gradient clipping, which explains why the performance is lower than previously reported.

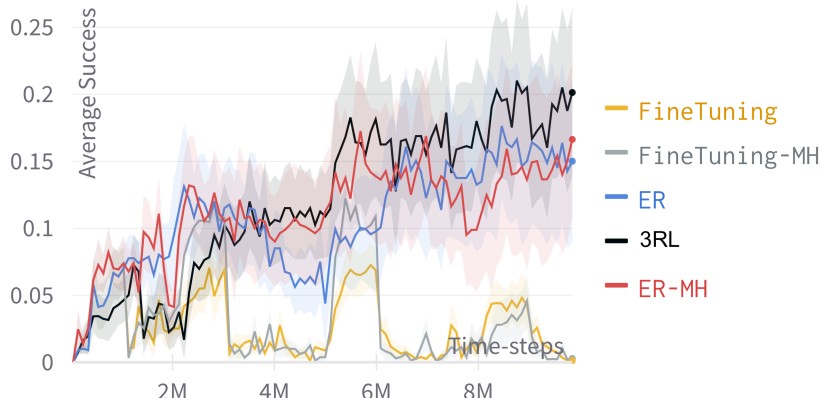

Figure 11: **CW10 experiment with larger neural networks.**. We repeated the CW10 experiment, this time with larger neural networks. Specifically, we added a third layer to the actor and critics. Its size is the same as the previous two, i.e. 400. The extra parameters have hindered the performance of all baselines. Note that it is not impossible that more well-suited hyperparameters could increase the performance of the bigger networks.

## J  Gradient conflict through time

See figure 12.

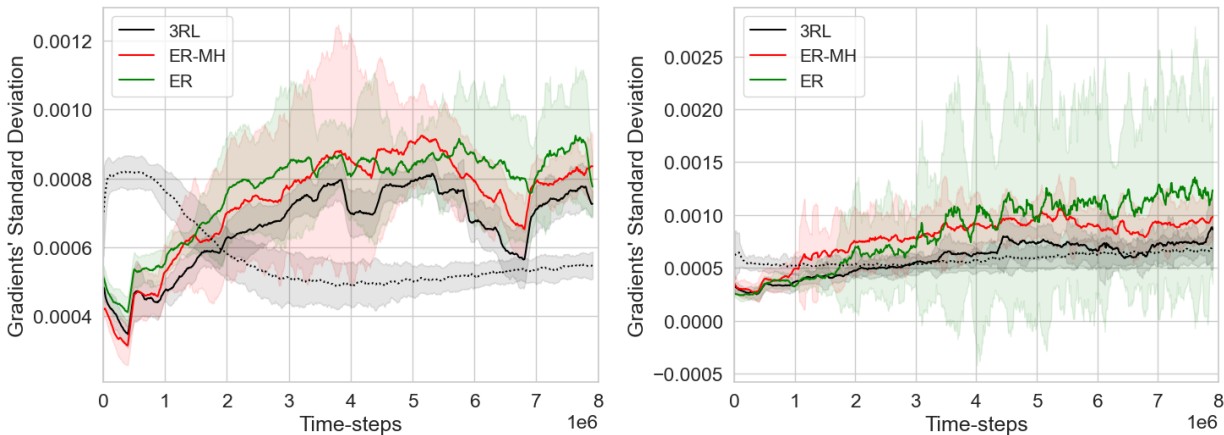

Figure 12: **Gradient variance analysis on CW20**. Comparison of the normalized standard deviation of the gradients for the actor (left) and critics (right) in for different CRL methods. For reference, we included MTL-RNN as the dotted line. The gradient alignment's rank is perfectly correlated with the performance rank.

