# OpenReview forum: "Task-Agnostic Continual Reinforcement Learning: In Praise of a Simple Baseline"
_TMLR — Rejected by TMLR_

### Review · Reviewer_SbPW · 2022-10-04

**Summary Of Contributions:**

This work studies the problem of task-agnostic continual reinforcement learning (TACRL) and the relations between TACRL methods in closely related problems of Multi-Task Learning (MTL) and Task-Aware Continual RL (CLR).

The paper makes the following core contributions:

- Summarize and contrast the core differences between the abovementioned problems and approaches for solving them.
- Propose a “previously unexplored and straightforward” baseline - replay-based recurrent RL (3RL) that augments an RL algorithm with recurrent NN (for addressing partial observability) and experience replay (for addressing catastrophic forgetting) in CL.
- Perform a large-scale comparison between 3RL and other model-baseline pairs on the popular continual learning benchmark Meta-World. The author show 3RL superior performance over existing baselines, even matching overcoming MTL and task-aware CRL soft upper bounds.
- The authors then lay 3 hypotheses for the superior performance of 3RL. After considerable attempts to verify these hypotheses, the authors discard 2 of them, while finding “some support” for the third hypothesis.

**Broader Impact Concerns:**

I have no concerns on the ethical implications of this work.

**Requested Changes:**

Please see the weaknesses above, especially regarding missing RNN-based baselines.

**Strengths And Weaknesses:**

## Strengths

- The paper takes one of the most important problems of continual learning: task agnostic setting. For me, the problem itself is real and practical.
- While adding an RNN to experience replay (ER) CRL approaches is not a big novelty in itself, the authors provide a thorough empirical evaluation and analysis to understand why this particular choice of model-baseline pair performs so well.
- I liked that the authors included an analysis of the hypotheses they lay regarding 3RL’s performance, even though they are later declined. While most papers typically don’t include such “negative results”, I believe readers would benefit from this information.

## Weaknesses

- I think the biggest weakness of the work is the lack of RNN-based baselines used in the comparison between approaches. It would be fair to include the combinations of baselines that also include RNNs. For example, the following baselines are missing
    - Finetunning-RNN
    - Finetuning-TaskID-RNN
    - ER-TaskID-RNN

    Without these baselines (especially the last one), it is difficult to conclude that “Task Agnosticism overcomes Task Awareness”. I know this is something that the authors consider, but I am not convinced that negative results on ER-RNN-MH justify not running all of these.

- The paper considers partial observability in continual RL stemming from task agnosticism. However, there is no discussion or empirical evaluation on environments where each task itself is substantially partially observable (IIRC there is not much partial observability in Meta-world). It would be interesting to empirically evaluate the findings of this work with and without partial observability of tasks themselves (as opposed to only partial observability stemming from task agnosticism).
- The authors mention using a single baseline for their empirical analysis as a limitation of this work. I agree with this and believe additional experiments on a different CRL benchmark would further convince the reviewers that the results are transferrable to other domains. The CORA benchmark [1] could be a suitable choice for this.

[1] Sam Powers, Eliot Xing, Eric Kolve, Roozbeh Mottaghi, Abhinav Gupta, “****CORA: Benchmarks, Baselines, and Metrics as a Platform for Continual Reinforcement Learning Agents”**** [https://arxiv.org/abs/2110.10067](https://arxiv.org/abs/2110.10067)

---

> ### Author Response · Authors · 2022-11-17
> **Reviewer SbPW Rebuttal**
>
> We thank the reviewer for his encouraging review.
>
> - I think the biggest weakness of the work is the lack of RNN-based baselines used in the comparison between approaches.
>
> We are still determining if running more mix-and-match baselines would add scientific value to the paper. The multi-head approach performed better than the TaskID one in all our experiments. We thus decided to run the most promising configuration, i.e., multi-head and RNN. We kindly recall that  each trial takes more than 4 days to run and a GPU per seed (and we try 8 seeds). Our empirical study is exhaustive, as suggested by uYqb and ox5E.
>
>
> - The paper considers partial observability in continual RL stemming from task agnosticism. However, there is no discussion or empirical evaluation on environments where each task itself is substantially partially observable (IIRC there is not much partial observability in Meta-world).
>
> Increasing the partial observability challenges of the benchmark is irrelevant to our message. In CRL, we focus on agents’ ability to accumulate skills and learn rapidly. [1] provides a throughout empirical study of recurrent RL being a strong baseline in all extensively-studied POMDPs. We do not believe repeating similar experiments would add much scientific value to our work.
>
> - The authors mention using a single baseline for their empirical analysis as a limitation of this work. I agree with this and believe additional experiments on a different CRL benchmark would further convince the reviewers that the results are transferrable to other domains. The CORA benchmark could be a suitable choice for this.
>
> We agree with the reviewer that more benchmarks would add scientific value to the paper. Nevertheless, we opted to spend all our computational budget on Meta World, which encompasses a broad task distribution and a realistically simulated Sawyer robot. In multi-task RL, we find that each benchmark requires expertise and so adding another one represents a significant investment.  As a point of comparison, multiple previously published works  report only Meta World experiments, e.g. [2,3,4,5]
>
> ----------------------
>
> [1] Recurrent Model-Free RL Can Be a Strong Baseline for Many POMDPs
>
> [2] Multi-Task Reinforcement Learning with Soft Modularization (NeurIPS 2020)
>
> [3] Continual World: A Robotic Benchmark For Continual Reinforcement Learning (NeurIPS 2021)
>
> [4] Gradient Surgery for Multi-Task Learning (NeurIPS 2020)
>
> [5] Multi-Task Reinforcement Learning with Context-based Representations (ICML 2021)

---

> > ### Comment · Reviewer_SbPW · 2022-11-25
> > **Thanks for the rebuttal**
> >
> > I thank the authors for the rebuttal.
> >
> > I'm still convinced that without above-mentioned baselines the experimental results are incomplete, especially if they only use a single benchmark. There is not even a big algorithmic novelty in this paper. In such cases, at least the experiments need to be VERY exhaustive which they aren't.
> >
> > > "Our empirical study is exhaustive, as suggested by uYqb and ox5E."
> >
> > As a reviewer I can see other's comments and discuss it with them directly (privately from you). There is no need to cite them under my review.
> >
> > If the authors maintain that they are unwilling to run the experiments, I'm going to recommend that the paper is rejected in its current form.

---

> > > ### Author Response · Authors · 2022-11-25
> > > **response**
> > >
> > > We are sorry for the confusion: we are not unwilling to run the aforementioned baselines. We simply wanted to open a discussion about what value they will bring to the paper. We will run the RNN baselines and tell you when the manuscript is updated. Thank you.

---

### Review · Reviewer_F3W2 · 2022-10-08

**Summary Of Contributions:**

The authors introduce 3RL, a simple baseline for task-agnostic continual reinforcement learning (TACRL). This consists an off-line reinforcement learning agent (SAC in their experiments) equipped with a replay buffer biased towards the current task and a recurrent network (GRU) for implicit task identification. They show that this method often matches multi-task learning (MTL) performance, which can be considered a soft performance upper bound due it utilizing all tasks at all times during learning. They also show that 3RL outperforms other approaches to TACRL and even CLR methods where task identity is known. The close by investigating the mechanism by which 3RL outperforms, and while inconclusive, settle on the idea that the recurrent network yields less gradient interference.

**Requested Changes:**

I'm afraid I believe a complete reworking of this paper is required to address my above concerns (i.e. different experiments with different methods). But in case I'm mistaken in that believe, here are some small-scale changes that would also be needed for acceptance.

1) Visualizations improvements. The cut in the y-axis of figure 5 is unacceptable. It's only purpose appears to be to magnify (IMO quite marginal) difference between 'independent RNN' and 3RL. Just have the axis uniformly go from 0 to 1. The number of curves also hurts readability here. I assume yellow and black are completely overlapping for the first task (which makes sense given they're related algorithms), but it could be that black has flatlined or overlaps with a different curve? I honestly can't tell even after zooming in on the PDF. Figure 6 could use some notion of spread, as it's hard to tell if the solid circles are significantly different. It's also unclear how how to interpret the size of the circles -- how does the variance of the Q-values relate to the hypotheses?

2) Bring hypothesis 4 to the main text and move hypothesis 2 to the appendix. Hypothesis 4 has a got more going on e.g. under additional data, other methods catch up in the main task setting. In hypothesis 2, you simply put forward a metric that doesn't pan out; I don't see this as particularly interesting.

3) Either cut Figure 1 bottom, or provide more evidence for its interpretation. E.g. Are the task initializations actually coming together in a meaningful way? Maybe they're staying the same, but merely subsequently moving around. Particularly since these are just low-dimensional projections of what we actually care about, I'm not confident anything substantial can be drawn from this figure.

**Strengths And Weaknesses:**

I appreciate the need for more papers like this: instead of proposing a new method, simply show that a reasonable (previously known) algorithm and architecture 'is all you need'. However, when this is the point, the burden is on the authors to show that their baseline is really performant compared to previously published methods in settings people care about. Unfortunately, I think this paper fails to pass that bar for several reasons.

1) No previously published methods are compared against. Most of the methods used weren't even designed for the TACLR setting (e.g. ER), and I believe the only one that was (ER-TAMH) hasn't been previously published elsewhere (or at least it isn't cited here), so it's poor performance isn't surprising. Indeed, from the method description (picking the most confident task head to act) I'm not surprised it doesn't work. Why not compare to any of the algorithms used in the Continual World paper (e.g. PackNet seems pretty strong)?

2) This environmental setup is, to the best of my knowledge, previously unpublished. The authors base their experiments on the setup described in Continual World, but utilize a harder version of Meta World, which can't seem find anyone one else using. This is problematic, as now it's impossible to directly compare to any previous work. This might be justified if you're exploring an area without in prior benchmarks, but here it's quite clear that you're wanting to compare to Continual World, so why not just use their exact setup?

3) Poor overall performance. The above issue is exacerbated by the poor overall performance of all methods considered (including 3RL). Only 4/10 tasks appear to ever get above chance performance. If you're trying to make the case that a baseline method is in some sense sufficient, you've got to either actually do well across the board do a very detailed hyper-parameter sweep to rule out the possibility that none of these methods sufficiently well-tuned to warrant any comparisons.

4) Other work on TACRL? I've gone through a lot of your references, but I only seem to find TAC-supervised or task-known-CLR.

5) It's unclear if even the previously published version of Continual World is broad enough to make the conclusions the authors want to make. All of the tasks are identifiable from a single frame, so they must only allow state features so as to increase task inference difficulty. All tasks are conceptually quite similar and only differ in their reward functions rather than their dynamics (e.g. changes to torque strengths). For example, why would we expect these conclusions to hold for e.g. continual learning of Atari games? I'll admit this is a high bar, but since the core contribution is highlighting the utility of a simple baseline, I view experimental variety as critical.

---

> ### Author Response · Authors · 2022-11-17
> **Reviewer F3W2 Rebuttal**
>
> We thank the reviewer for the thorough recommendations to improve the paper.
>
> - No previously published methods are compared against.
>
> This work shows (to our surprise) that a simple memory mechanism can be used to improve the performance of methods without the need to add other continual-learning-specific mechanisms (e.g. for task inference or not forgetting). This is orthogonal and questions the need for these mechanisms proposed by previous works, including PackNet. As such, we do not see the value of adding these baselines.
>
> - This environmental setup is, to the best of my knowledge, previously unpublished.
>
> This work tackles Task-agnostic Continual RL. Hence, comparing directly with Continual World, a task-aware benchmark is not an option. Nevertheless, our task-aware baselines are comparable to Continual World’s methods.
>
> - Poor overall performance.
>
> The performance discrepancy comes from the switch from Meta-World v1 to v2, explained in the Appendix. See Figure 7, which validates our SAC implementation, and Figure 8, showing that single task performance in CW10 v2 is below 50%, much lower than the reported performances in CW10 v1.
>
> - Other work on TACRL?
>
> A non-exhaustive list of TACRL work can be found in our related work:
>
> "Closer to our training regimes, TACRL is an actively studied field. Xu et al. (2020) uses a infinite mixture of Gaussian Processes to learn a task-agnostic policy. Kessler et al. (2021) learns multiples policies and casts policy retrieval as a multi-arm bandit problem. As for Berseth et al. (2021); Nagabandi et al. (2019), they use meta-learning to tackle the task-agnosticism part of the problem.”
>
> Further TACRL works are found in [1] (see below).
>
> - All tasks are quite similar and only differ in their reward functions.
>
> In Meta-World, tasks must share some similarities (fixed physics/dynamics combined with a rich shared reward structured). But the Meta-World task distribution is broader than OpenAI's Mujoco, the typical environments found in continual RL papers,  tasks which are much more similar (e.g. a Half-Chettah setting that only changes friction).
>
> Furthermore, fixing the dynamics and focusing on tasks sharing a common reward structure has been extensively studied in all papers strictly focusing on Meta-World, e.g. [2,3,4,5,6] as well as others, e.g. [7,8,9]
>
> - why would we expect these conclusions to hold e.g., continual learning of Atari games?
>
> We are making no claims that our findings generalize to video games. As a matter of fact, we are trying to get away from artificial CRL benchmarks. We do not see why our conclusion should extend to less practical settings like video games. Furthermore, Atari can not be formulated as task-agnostic CRL: there’s no possible hidden task that needs to be inferred.
>
> - Visualizations improvements.
>
> We thank the reviewer for their comments on improving Figure 5. We are working on it.
>
> - Bring hypothesis 4 to the main text and move hypothesis 2 to the appendix.
>
> We further thank the reviewer for this suggestion. We will bring hypothesis #4 to the main text. We are, however, of the opinion that hypothesis #2 should remain in the main text, as its invalidation increases the support for the other hypotheses.
>
> ----------
>
> [1] Towards Continual Reinforcement Learning: A Review and Perspectives
>
> [2] Multi-Task Reinforcement Learning with Soft Modularization
>
> [3] Meta-World: A Benchmark and Evaluation for Multi-Task and Meta Reinforcement Learning
>
> [4] Continual World: A Robotic Benchmark For Continual Reinforcement Learning
>
> [5] Gradient Surgery for Multi-Task Learning
>
> [6] Multi-Task Reinforcement Learning with Context-based Representations
>
> [7] Successor features for transfer in reinforcement learning
>
> [8] Same State, Different Task: Continual Reinforcement Learning without Interference
>
> [9] CoMPS: Continual Meta Policy Search

---

### Review · Reviewer_ox5E · 2022-11-03

**Summary Of Contributions:**

This paper explores the observation that the 3RL baseline (i.e. an RL agent with an experience replay buffer and a recurrent network), outperforms many algorithms manufactured for the Task-agnostic continual reinforcement learning problem specification. This paper explores three hypotheses (and one hypothesis in the appendix) to explain this observation:
1. the recurrent network individually improves on a single task,
2. the recurrent network increases "parameter stability",
3. the recurrent network is more able to remember old tasks and ground new ones in previous examples.

The goal of the paper is to only explore this baseline, comparing to more specialized architectures.

**Requested Changes:**

Address the comments above.

**Strengths And Weaknesses:**


**Strengths**

This paper generally is really well done, written, and presents a very clear message. I also appreciate the clarity the authors brought in terms of their empirical evaluation and hypothesis testing. I am not a part of the discrete task continual learning community so my general knowledge of background is not as deep, I will thus defer my judgement on novelty of this observation to other reviewers/later discussion. I also appreciate the limitations being well laid out in section 6.

Below I have some concerns/questions which are numbered.

**Weaknesses**
1. I think the justification for using the previously chosen hyperpramaters is weak, and not well justified. While I think it is ok to use a shared hyperparameter configuration for all your models, this should be better justified and explained (especially listing exactly the parameters that are shared amongst the models and those that are only used by some if they exist). I also am unclear if the hyperparameters were chosen originally for MW v1 or MW v2. *Some other quibbles:*
    - What is the truncation length used for backpropagation through time? Or did you use entire episodes? How was the hidden state initialized if whole episodes weren't used?
2. When making the models bigger, using the same hyperparameters of the smaller networks is not really valid as an experiment. While it is computationally expensive, to draw any kind of conclusions about the larger models a better sweep is needed. For these experiment, reporting the actual size of all the used networks (say in # of tunable parameters) would provide much more insight into the discrepancies you are discussing.
3. If you can get more runs of your experiments, this would help substantially. 8 seeds for any kind of t-testing is quite low. Another idea would be to plot all the individual learning curves so we can see the spread that is hidden by the average.
4. I'm unconvinced by the entropy-like metric to compute the parameter stability of your network, especially when we average over these over all the parameters. One problem I could imagine is some parameters are stable, and some aren't stable. What the RNN could be facing (although I'm doubtful if this hypothesis is true) is it has parts of the network which are stable and parts which are not. I think digging a bit deeper beyond the expected value of the entorpy here would be worthwhile (i.e. look at indv weight entropies).
5. I think the conversation in appendix J should be in the main text, as it justifies your metric for gradient conflict (which differs from Yu et al). '



**Minor Quibbles:**
- "limitation of standard learning agents": while I agree this is a problem shared by many agents with deep learning at their core, I'm not sure it is standard across all standard agents. I would modify to note that this is especially the case in DRL.
- In the appendix, many of the citations don't have parenthesis when they should . Please go through and edit these.
- Page 9: "However, now the agent needs to subsequently push and not pull, *has* was required..."

---

> ### Author Response · Authors · 2022-11-17
> **Reviewer ox5E Rebuttal**
>
> We thank the reviewer for taking the time to lay out recommendations to improve the paper.
>
> - I think the justification for using the previously chosen hyperpramaters is weak, and not well justified.
>
> The choice of hyperparameters required quite some work. Initially, we used the hyperparameters prescribed by Continual World without any success. We then tried the MetaWorld v2 prescribed hyperparameter, which helped us match MetaWorld’s multi-task reported results. However, the continual learning baselines would still suffer from largely unstable training. After further experimentation, we observed that gradient clipping could stabilize training, and that clipping the gradients to a norm of 1 achieved the desired behavior across all methods. We will update the Appendix to explain this and provide the exhaustive list of hyperparameters.
>
> Running a hyperparameter search for all individual methods was outside our computational budget.
>
> The truncation length was 15. We did not try any other values. [1] suggest that searching for the correct truncation value would further increase the outperformance of 3RL over the other baselines. The initial hidden state is learned: the RNN ingests an empty sequence of states, actions, and rewards of length 15 and outputs a learned initial hidden state.
>
> - When making the models bigger, using the same hyperparameters of the smaller networks is not really valid as an experiment.
>
> We find this to be a valid inquiry and is why we left this finding in the Appendix. We added a word of caution. We further reported the number of parameters each method enjoys.
>
> - If you can get more runs of your experiments, this would help substantially.
>
> Running more than eight seeds per configuration was outside our computational budget.
> Furthermore, previous work tend to operate in the same regimes, e.g. 4 seeds in [1], 10 seeds in the original Meta-World paper [2], 6 seeds in [3].
>
> Reproducing our reported experiments requires 4 GPU years, an order of magnitude less than what we spent.
>
> - I'm unconvinced by the entropy-like metric to compute the parameter stability of your network, especially when we average over these over all the parameters
>
> We disagree that the \emph{average} entropy is not a good metric for what we want to evaluate. The reviewer mentions that it might be a problem if, e.g., half the parameters are stable and half are not. We do not see any problem with this. If all parameters are stable, there is no plasticity to learn new tasks.
> The proposed metric takes all its sense when used relatively, (i.e. to compare a model relative to another) and not as an absolute measure of parameter stability. E.g., the metric might have shown that 3RL enjoyed more stable parameters than the baselines, which suffer from more forgetting. This would have supported hypothesis #2 (RNN increases parameter stability, thus decreasing forgetting).
>
> - I think the conversation in Appendix J should be in the main text, as it justifies your metric for gradient conflict (which differs from Yu et al).
>
> We agree with this suggestion and will move this conversation to the main text. Thank you!
>
> - limitation of standard learning agents
>
> Indeed, this is mostly a limitation of deep learning agents. We have updated the manuscript.
>
> We thank the reviewer for pointing out the wrong citation type in the Appendix and the typo.
>
> -----------
>
> [1] DisCor: Corrective Feedback in Reinforcement Learning via Distribution Correction
>
> [2] Meta-World: A Benchmark and Evaluation for Multi-Task and Meta Reinforcement Learning
>
> [3] CoMPS: Continual Meta Policy Search

---

### Review · Reviewer_uYqb · 2022-11-05

**Summary Of Contributions:**

The paper proposes a simple baseline for multi-task continual learning, using an RNN based model across tasks (in the multi-task and continual learning settings). They find that with the recurrent memory, this agent can outperform task-aware agents. Further, if the agent is able to replay trajectories from previous tasks (in the continual learning setting), it can get close to multi-task RL performance. The authors present results on the challenging meta-world and continual world environments.

**Requested Changes:**

Some explanation/discussion for using parameter entropy as a proxy for measuring forgetting could help strengthen that section of the experiments.

**Strengths And Weaknesses:**

Strengths

1. Significance - The problem settings considered in this paper - that of multi-task learning and continual learning, are quite significant and important to the robot learning community. The authors include analysis on meta-world and continual world, which have are established benchmarks in the community which are often not used by many papers due the difficulty of the tasks involved, but the environments are closer to the eventual real world robotics settings (since they involve a sawyer robot that needs to manipulate different objects). Thoroughly evaluating different common approaches, as well as the simple RNN based approach here will help the community evaluate the utility of new novel algorithms that seek to address similar problems.

2. Comparison to task aware methods - The results where the proposed baseline outperforms task-aware methods (where the task is provided via some fixed context) are very interesting. Providing explicit task-ids is not feasible in actual open-world learning (since it's unclear where the task boundaries lie), and showing that a task agnostic approach can actually learn similarities/differences between different tasks in a multi-task RL setup is encouraging. This makes an even stronger case for new work in the community to compare against the proposed approach.

Weaknesses

1. Parameter stability analysis & relation to forgetting - The authors include some experiments where they argue that increased parameter stability leads to less forgetting, and include an entropy like metric to measure the change in the parameters. It is unclear how well changes in the parameter value correspond to the policy forgetting since the same behavior could be encoded differently and the same parameter values might lead to different behavior for different inputs.

---

> ### Author Response · Authors · 2022-11-17
> **Reviewer uYqb Rebuttal**
>
> We thank the reviewer for their favourable review. Below we provide replies to each of their comment.
>
> “Parameter stability analysis & relation to forgetting”
>
> We clarify that we do not propose parameter entropy as a proxy for measuring forgetting. Previous work shows that methods that can successfully alleviate catastrophic forgetting have stable parameters. This can be achieved in a hard way, e.g., PackNet, or in a soft way, e.g., EWC. Accordingly, we test the hypothesis that 3RL achieves the same result through a similar behaviour, i.e., parametric stability.
>
> It is true, however, that the 3RL could be stable in its function but not in its parameters. We have
> updated the manuscript to reflect this.

---

### Author Response · Authors · 2022-11-17
**General reply**

We want to thank the reviewers. We appreciate that all reviewers have read our work thoroughly and taken the time to write constructive reviews. We are encouraged that the reviewers found the tackled setting important (uYqb, SbPW) and that we are setting a strong baseline for future work (uYqb), that we need more papers like this (F3W2), that our empirical study was thorough/exhaustive (uYqb, ox5E, SbPW), that the findings are very interesting (SbPW), that the paper generally is really well done, written, and presents a very clear message (ox5E) and that the hypothesis testing was appreciated (ox5E, SbPW).

We are currently rewriting the paper in light of constructive criticisms. E.g., we agree that Figure 5 could be better (F3W2) and are working on a better version. We will move Hypothesis #4  to the main text (F3W2). We will further justify and clarify our choice of hyperparameters (ox5E). We will move the justification for the gradient conflict metric in the main text (ox5E). A revised version will be available before the end of next week.

All concerns are addressed with their respective reviewers in the replies.

---

### Author Response · Authors · 2022-11-26
**revisions**

We want to thank the reviewers again for their constructive criticisms. We revised the manuscript in light of the recommendations. The main revisions are:

- We updated and moved into the main text Hypothesis #4 (now Hypothesis #3) (F3W2)
- We removed Figure 5’s cut-off and improved readability (F3W2)
- We moved the gradient conflict metric argumentation in the main text (ox5E)
- We clarified the parameter stability analysis & relation to forgetting (uYqb)
- We clarified how the entropy-like metric to compute the parameter stability should be interpreted (ox5E)
- We added a justification for our hyperparameters as well as a hyperparameter table (see App. D.1) (ox5E)

All revisions are highlighted in blue.

We believe the manuscript's quality has significantly improved and welcome any further recommendations.

---

### Decision · Action_Editors · 2022-11-30

**Recommendation:** Reject

**Comment:**

The reviewers all had a lot of sympathy for this paper, and would like to see it published in the long term. There were, however, concerns from multiple reviewers about the basis of comparison and lack of comparable baselines. Reviewer F3W2, in particular, points out a number of comparable works which are referenced in the paper but not compared against. The authors point out that their experimental setup does not allow comparison, but then why not examine the performance of this baseline in the related CW experimental setup where other baselines do apply? The reviewers discussed this point, and after the rebuttal determined that the outstanding issue of proper comparison still needed addressing, and still warrants a resubmission after revision to properly examine the paper anew. Furthermore, there needs to be more rigour in the examination of hyperparameters and the depth of the ablation study, for readers to truly have confidence in the results.

**Audience:**

This paper is relevant to TMLR's audience, without question. If strengthened by addressing the criticism offered by the reviewers, we have no doubt it will be highly interesting.

**Claims And Evidence:**

This work presents evidence that a simple baseline works better than more complex methods in a continual task-agnostic setting. It gives empirical evidence for this argument which is clear and not unconvincing, but lacks adequate comparisons, and is open to the objection that the setting itself is a variation on existing work where comparison could have been drawn which is better framed by the existing literature.